# In-situ observation of trapped carriers in organic metal halide perovskite films with ultra-fast temporal and ultra-high energetic resolutions

Kanishka Kobbekaduwa [1], Shreetu Shrestha[2], Pan Adhikari [1], Exian Liu [1], Lawrence Coleman [1], Jianbing Zhang [3], Ying Shi[4], Yuanyuan Zhou [5], Yehonadav Bekenstein[6], Feng Yan [7], Apparao M. Rao [1], Hsinhan Tsai [2], Matthew C. Beard[8], Wanyi Nie [2]✉ & Jianbo Gao [1]✉

We in-situ observe the ultrafast dynamics of trapped carriers in organic methyl ammonium lead halide perovskite thin films by ultrafast photocurrent spectroscopy with a sub-25 picosecond time resolution. Upon ultrafast laser excitation, trapped carriers follow a phonon assisted tunneling mechanism and a hopping transport mechanism along ultra-shallow to shallow trap states ranging from 1.72–11.51 millielectronvolts and is demonstrated by time-dependent and independent activation energies. Using temperature as an energetic ruler, we map trap states with ultra-high energy resolution down to < 0.01 millielectronvolt. In addition to carrier mobility of ~4 cm$^2$V$^{-1}$s$^{-1}$ and lifetime of ~1 nanosecond, we validate the above transport mechanisms by highlighting trap state dynamics, including trapping rates, de-trapping rates and trap properties, such as trap density, trap levels, and capture-cross sections. In this work we establish a foundation for trap dynamics in high defect-tolerant perovskites with ultra-fast temporal and ultra-high energetic resolution.

[1] Department of Physics and Astronomy, Ultrafast Photophysics of Quantum Devices Laboratory, Clemson University, Clemson, SC, USA. [2] Center for Integrated Nanotechnology, Los Alamos National Laboratory, Los Alamos, NM, USA. [3] School of Optical and Electronic Information, Huazhong University of Science and Technology, Wuhan, People's Republic of China. [4] Institute of Atomic and Molecular Physics, Jilin Provincial Key Laboratory of Applied Atomic and Molecular Spectroscopy, Jilin University, Changchun, People's Republic of China. [5] Department of Physics, Hong Kong Baptist University, Kowloon Tong Hong Kong, People's Republic of China. [6] Department of Materials Science and Engineering, Technion, Haifa, Israel. [7] Department of Metallurgical and Materials Engineering, The University of Alabama, Tuscaloosa, AL, USA. [8] National Renewable Energy Laboratory, Golden, CO, USA. ✉email: wanyi@lanl.gov; jianbogao.nano@gmail.com

**B**uilding a near ideal optoelectronic or photonic device requires mapping a full-carrier generation-decay process, including the transport, recombination, and trap dynamics. In view of this, trap (or defect) states and their twin "doping state" play key roles in manipulating electrical and optical properties in materials science and solid-state physics. Hence, understanding carrier dynamics of trap states is vital for materials science and their applications in optoelectronic or photonic devices. This is particularly important and urgent for organic metal halide perovskites, which contain high densities of shallow trap states that undermine their otherwise superior device performance, since the large trap state density is a consequence of solution-processed fabrication of these materials under low temperatures[1].

Although the power conversion efficiency (PCE) of perovskite solar cells has soared to >25% in the last decade[2], the PCE can be further improved by reducing the recombination processes from these trap states. Defect states have an impact on solar cell stability and efficiency. Overcoming this impact is a major challenge in commercialization of this solution-processing technology. In addition, these high defect-tolerant materials have applications in other optoelectronic devices, such as light-emitting diodes (LED)[3,4], lasers[5,6], photodetectors[7,8], and X-ray detectors[9,10]. Despite all the tremendous work focused on the correlation between the trap (or defect) states and perovskite film structure characteristics (such as surface morphology, domain size, and grain boundaries), the dynamics of trapped carriers in high densities of shallow trap states still remains unclear[11,12]. Thus, in situ observation of trapping dynamics is of paramount importance in the evolution of perovskite optoelectronics and photonics.

The most popular approaches to studying trapped carrier dynamics are transient absorption spectroscopy (TAS)[13,14] or similar pump-probe approaches[15,16] and time-resolved photo-luminescence (TRPL)[13,17,18]. Despite the high temporal resolution (down to a picosecond (ps) time scale for TAS), ultrafast optical spectroscopies are incapable of directly collecting carriers as current in situ optoelectronic devices because they operate under a zero electric field environment. The underlying photo-physics of traditional ultrafast optical spectroscopies can only probe carriers localized in regions of a couple of nanometers. Thus, they are incapable of long-range charge carrier transport studies due to the absence of an external electric field, which provides the momentum to push charge carriers. While new transient absorption imaging techniques[19,20] have proven to be powerful tools that can probe long-range diffusion dynamics through analysis of photon emission at electron–hole recombination, the long-range measurements are primarily focused on excitons that are charge neutral electron–hole pairs that do not carry current, thus aren't useful in charge carrier transport studies required for in situ devices. In view of this, there is a significant gap between the study of dynamics in ex situ materials and the dynamics of in situ devices. Equally important, they lack the required energetic resolution down to millielectronvolts (meV) to reveal the shallow trap states (in the range of 100 s of meVs) due to the low energy resolution of the probe beam, while the detection of ultra-shallow trap states, which are in the range of a couple of meVs is even harder.

Although the time-of-flight photoconductivity[21] and photo-charge extraction by linearly increasing voltage (photo-CELIV)[22] techniques are capable of studying trapped carrier dynamics in situ, and shallow trap states can be mapped by varying the temperature, the temporal resolution is in the nanosecond (ns) to microsecond (μs) range due to device architecture. Hence, these techniques are ideal for characterizing deeper trap levels to which carriers take an extended time period to decay, such as those existing in the bulk of single-crystalline perovskites, as described by Musiienko et al.[23,24] However, for accurate measurements, these techniques require large device cross sections, resulting in greater device RC constants that lead to device rise times in the order of ~10 ns (see Supplementary Methods). This results in the omission of the ps–ns region in the photocurrent decay thus, these techniques are "blind" to shallow trap dynamics which happen near the edge of the conduction band (CB) and occur in this ps–ns time scale or even less. Therefore, we have a "temporal blind spot" in the ps to ns time range with respect to in situ devices, which hinders the characterization of shallow defects. In view of all these constraints, this work aims bridge the following gaps in knowledge: (i) ultrafast carrier dynamics between ex situ perovskite materials and in situ device carrier dynamics; (ii) link the ps time resolution in the optical approach and the ns time resolution response in the device approach; and (iii) reveal carrier dynamics using temperature as a ruler and measure trap states with ultrahigh energy resolution.

Here, we focus on an organic–inorganic metal halide perovskite material, defect-filled methyl ammonium lead iodide (MAPbI$_3$) perovskite system[25,26] to understand the carrier trapping dynamics process, using an ultrafast photocurrent spectroscopy (UPCS) technique with a time resolution of sub-25 ps. We discovered that upon ultrafast laser excitation, trapped carriers follow two transport mechanisms; (1) a phonon-assisted tunneling (PAT) mechanism from 25 to ~220 ps, described along ultra-shallow trap states where carriers move from the trap states to the lower conduction band (CB) by tunneling through the energy barrier via phonon absorption (~1.72–10.15 meV) and demonstrated by time-dependent activation energies; (2) a hopping mechanism within shallow trap states (>10.15 meV) after ~220 ps, supported by time-independent activation energies. These activation energies are found by using temperature as a gauge to obtain an ultrahigh energy resolution that cannot be achieved using other techniques. The observed mechanisms are further validated by the correlation among the trap state properties, indicating that the capture cross section decreases from ~10$^{-8}$ cm (1 Å) to ~10$^{-9}$ cm (0.1 Å), leading to the decreasing of trapping probability. This is consistent with the decrease of trapping rates from ~10$^{11}$ to ~10$^9$ s$^{-1}$, as well as the decrease in de-trapping rates. In addition, we obtain values for carrier mobility (minimum estimate of ~4 cm$^2$ V$^{-1}$ s$^{-1}$ for a quantum yield of 100 %) and carrier lifetime (~1 ns as seen in Supplementary Fig. 1). The UPCS experimental setup is demonstrated in Fig. 1 and is superior in temporal, energy resolution, and in situ measurement ability to previously reported ultrafast techniques in other semiconductor systems utilizing organic semiconductors[27,28], nanocrystal

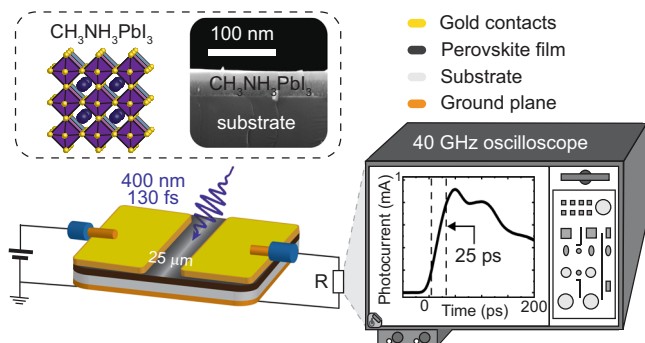

**Fig. 1 Experimental schematic setup of UPCS.** The thin film perovskite photoconductive device is integrated into a microstrip transmission line structure to have a sub-25 ps time resolution ultrafast photocurrent, which is collected by a sampling oscilloscope. The ultrafast laser pulse duration is 130 fs. The top left panel is the MAPbI$_3$ lattice structure and thin film cross section SEM image on glass substrate.

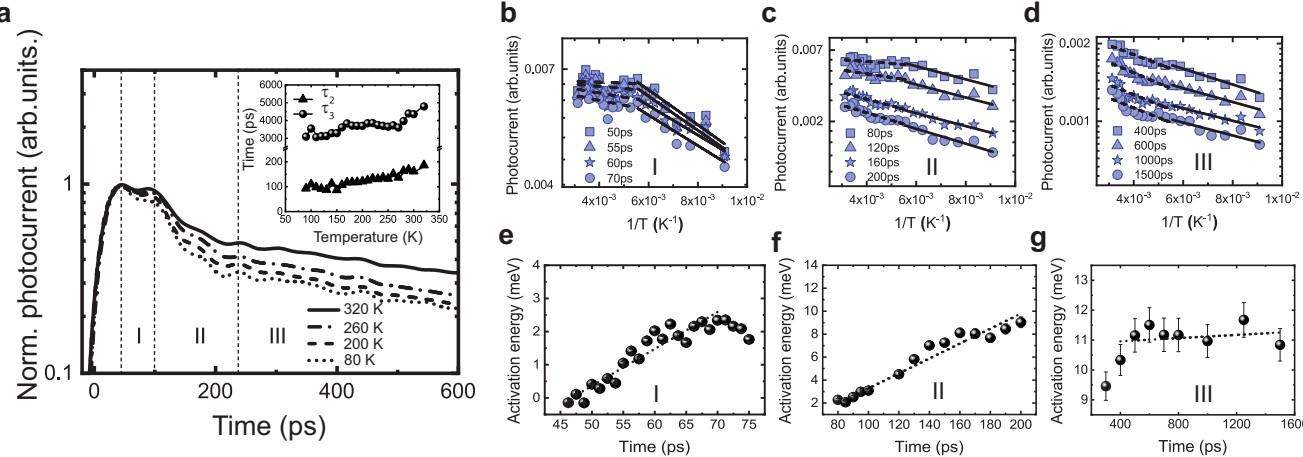

**Fig. 2 Ultrafast photocurrent dependence on temperature. a** The normalized ultrafast photocurrent decay at various temperatures. The inset is decay constants dependence with temperature of region II and region III. Arrhenius plots at various times of region I (**b**), region II (**c**), and region III (**d**). **e–g** The corresponding thermal activation energy dependence in the 180–320 K temperature range with time.

semiconductors[29], and 2D layered semiconductors[30–32]. In contrast to ultrafast optical spectroscopies, the UPCS studies the ultrafast carrier dynamics in situ devices by directly collecting carriers as a current, under diverse tunable experimental conditions, such as temperature, excitation photon energy, photon density under linear and nonlinear excitations, and an electrical field. Using temperature as a gauge, this technique enables us to measure the trap states at an ultrahigh resolution. In addition, to trapping dynamics, UPCS can also be employed to study carrier photogeneration, transport, and recombination processes.

In this work, a MAPbI$_3$ thin film photoconductive device is integrated into a microstrip transmission line architecture to reduce the device RC time constant (see Supplementary Methods for detail device fabrication, experimental setup, and materials synthesis). To avoid silver, aluminum, or other contact degradation of perovskites, gold (Au) transmission line contacts are pre-patterned on the perovskite thin films[33,34]. The ultrafast photocurrent generated upon an ultrafast laser excitation (400 nm, 130 fs pulse laser) is collected by a high-bandwidth sampling oscilloscope, while the laser illuminates whole device, including bulk perovskite film and Au/perovskite interfaces (shown in Supplementary Fig. 2). As a result, charge carriers that contribute to the ultrafast photocurrent is mainly from the bulk of the film. This is further demonstrated by the similar photocurrent decay trend when laser illuminates from either bottom or top side of the film (Supplementary Fig. 3). The achieved time resolution defined by the period between 10% and 90% of photocurrent peak is ~25 ps, which is limited by the bandwidth of the sampling oscilloscope, coaxial cables, and the photoconductive device impedance. The ohmic nature of the contacts is shown by the dark IV given in Supplementary Fig. 4.

## Results

**Temperature dependence of ultrafast photocurrent.** Figure 2a shows the normalized ultrafast photocurrent at various temperatures under an electric field of $4 \times 10^3$ V cm$^{-1}$ and 20 μJ cm$^{-2}$ laser flux. The excitation photon flux is kept in the linear excitation region to relate the carrier dynamics with the present understanding of similar solar cell operation conditions. The highlights from observations made through multiple devices (see Supplementary Fig. 5 for another typical device) can be summarized in the following statements: (1) it is evident that there are three distinct decay regions, marked by I, II, and III. Beginning from the photocurrent peak edge ~25 to ~100 ps (region I), followed by a short interim

region from 100 to 240 ps (region II), and a longer decay process after 240 ps (region III); (2) all three regions show thermally activated behavior as the photocurrent decays faster with decreasing temperature confirmed through the temperature dependence of the lifetime constants in regions II and III, $\tau_2$ and $\tau_3$, respectively, illustrated in the inset of Fig. 2a; (3) the decrease of photocurrent with decreasing temperature can be represented by an Arrhenius relation $\sim e^{-\frac{\Delta E}{k_B T}}$, where $\Delta E$ is the activation energy, and $k_B$ is the Boltzmann constant. Accordingly, Fig. 2b–d shows the photocurrent dependence with temperature at different times in all three regions; (4) the corresponding derived time-dependent and independent activation energies in the 180–320 K temperature range (the dashed lines in Fig. 2b–d) shown in Fig. 2e–g, and the time-independent activation energies in the 80–80 K temperature range (the full lines in Fig. 2b–d) shown in Supplementary Fig. 6 demonstrate the carrier dynamics in an energy landscape. In region I, the activation energy increases steadily with time from 0.81 to 2.51 meV, while it increases rapidly from 2.51 to 8.52 meV in region II. However, an activation energy level of 11.01 meV is a near constant in region III at larger time values.

The trap levels identified in this work are far lower than from other optical and electrical methodologies by TRPL[35], time-resolved microwave conductivity technique[36], photothermal deflection spectroscopy (PDS), and Fourier transform photocurrent spectroscopy (FTPS)[37]. This difference is due to the lower temporal and energetic resolutions of above techniques, particularly in the ns–μs range, that impede the measurement of ultrashallow traps. The existence of ultra-shallow traps leads to ultrafast trapping events. These traps aid in faster transport of charge carriers due to their proximity to the CBM, ease of de-trapping into the CBM, and lowering of the dominant non-radiative recombination processes in organic lead halide perovskites[38,39], which will result in typically lower carrier transport times and higher device efficiency compared to deeper and slower trapping events. In solar cells, this can lead to an increase of the open-circuit voltage ($V_{oc}$) from carriers that can be extracted before trap-assisted recombination occurs[40]. The distinctive use of temperature as a ruler for energy measurement is one of the unique features for the ultra-shallow trap levels revealed by the UPCS. We also identify the trap densities to be on the order of $10^{15}$ cm$^{-3}$, which is consistent with shallow trap density values obtained by other techniques such as space-charge limiting current[41], optical methods, such as TRPL[42], TAS[43], and from several other methods[1,44,45] as well.

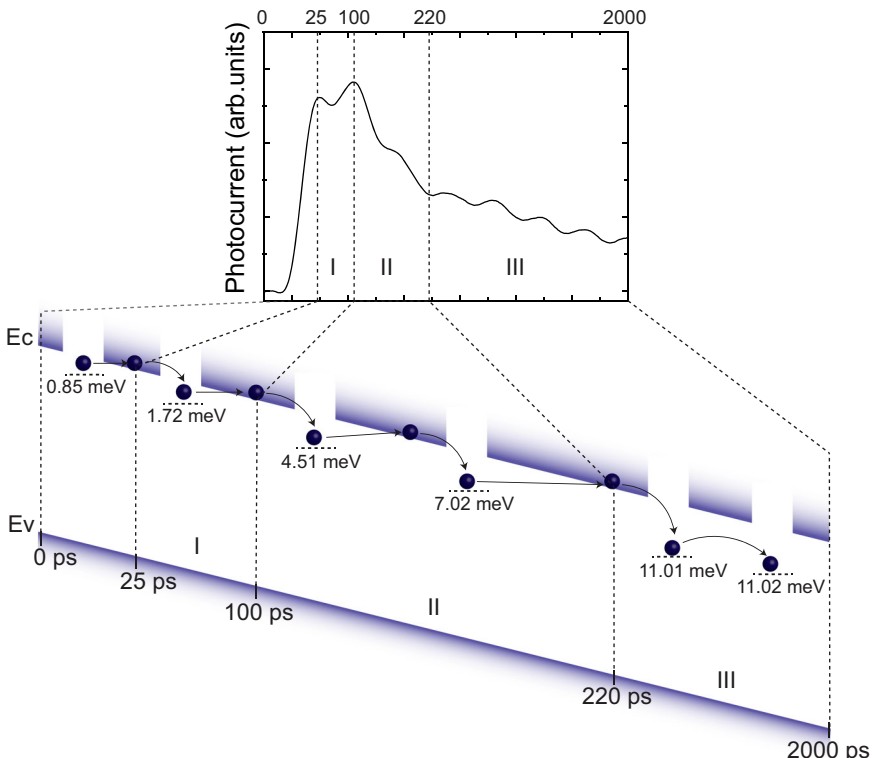

**Fig. 3 The schematic of phonon-assisted tunneling and hopping transport dynamics.** Charge carriers that initially fall into ultra-shallow trap states are; (I) de-trapped on to the conduction band ($E_c$) via tunneling before slowly recombining, (II) relax into slightly deeper ultra-shallow states and de-trapped again, (III) "hop" between shallow trap states. Only electron motion is demonstrated in the model since holes follow the same mechanism. The zero-time point is set at 10% of maximum photocurrent. Note: to clarify the transport process and for convenience only some of the typical states existing within each region are shown. Whereas in the actual case trap levels in the regions between 0 and 2000 ps are continuous as seen in Fig. 2. The valance band is depicted by $E_v$.

**Transport mechanism.** To describe the transport between the time-dependent and time-independent thermal activation energies in these three temporal regions, we use a PAT mechanism followed by a hopping mechanism, as shown in Fig. 3. In particular, we refer the trap states under 10.15 meV and above 10.15 meV as ultra-shallow and shallow, respectively. Initially, the photogenerated carriers relax to the band edge in <1 ps, then fall into ultra-shallow trap states in <25 ps. Although we are yet to achieve a rise time resolution to resolve the ultrafast carrier dynamics prior to 25 ps, the initial photocurrent peak can be attributed to carriers falling into the trap state, as indicated by the slightly increasing photocurrent with increasing temperature (Supplementary Fig. 7). Then, in region I from 25 to 100 ps and region II from 100 to 220 ps, carriers follow the PAT transport mechanism in ultra-shallow trap states, where the carriers' tunnel through the energy barrier (now reduced by the applied electric field) between the ultra-shallow trap states and the CB; this is confirmed by the activation energy increase with time. For instance, the thermal activation increases from 1.01 meV at 55 ps to 8.53 meV at 200 ps. Similar mechanisms have been reported in other low-carrier mobility semiconductor systems, including amorphous selenium[46] and organic amorphous polymers[47,48]. In region III, the carriers will hop among the shallow trap states until they fall into slightly deep trap states, indicating a time-independent thermal activation. Although the carrier demonstrated is an electron, a similar mechanism can be applied to holes as well. The PAT and later the hopping transport mechanisms are also consistent with the calculated lowest estimate mobility from the photocurrent peak of ~4 cm$^2$ V$^{-1}$ s$^{-1}$ (see Supplementary Note 1).

**The effect of a variable electric field.** The carrier dynamics representation is analyzed through the ultrafast photocurrent dependence with a variable electric field (Fig. 4). As seen in the inset of Fig. 4b, in the range 0–4 × 10$^3$ V cm$^{-1}$ the decay time constants $\tau_2$ and $\tau_3$ increase with increasing external electric field; this is due to the increase in carrier de-trapping effect since the trapped carriers in ultra-shallow trap states can be depopulated back to CB via PAT by the lowering of the CB due the applied electric field. The PAT model is further confirmed through the ultrafast photocurrent dependence with electric field at a low temperature of 180 K shown in Fig. 4c, d, where due to sparse phonon interactions the de-trapping process is suppressed even at larger electric fields (up to 4 × 10$^3$ V cm$^{-1}$). In the absence of large thermal energy, the decay time constants $\tau_2$ and $\tau_3$ are lower, in contrast to that of the high temperature (300 K) measurements. Further increase of the electric field (larger than 4 × 10$^3$ V cm$^{-1}$) at high temperatures results in the decrease of decay time, which is contradictory to the de-trapping mechanism. This unusual behavior is attributed to the onset of ion transport[49], which induces more recombination events. Although ion migration is ever present with the application of an external electric field, Fig. 4b, d indicates the screening effects due to ion migration will begin to influence the free flow of charge carriers (photocurrent), when the external electric field is larger than 4 × 10$^3$ V cm$^{-1}$ as the accumulation of ions on the perovskite–gold interface will result in the formation of an internal electric field that directionally opposes the externally applied electric field, reducing the flow of charge carriers. The observed reduction of $\tau_2$ and $\tau_3$ (insets of Fig. 4b, d) is indicative of this process, where the rate of recombination becomes larger at those electric fields as

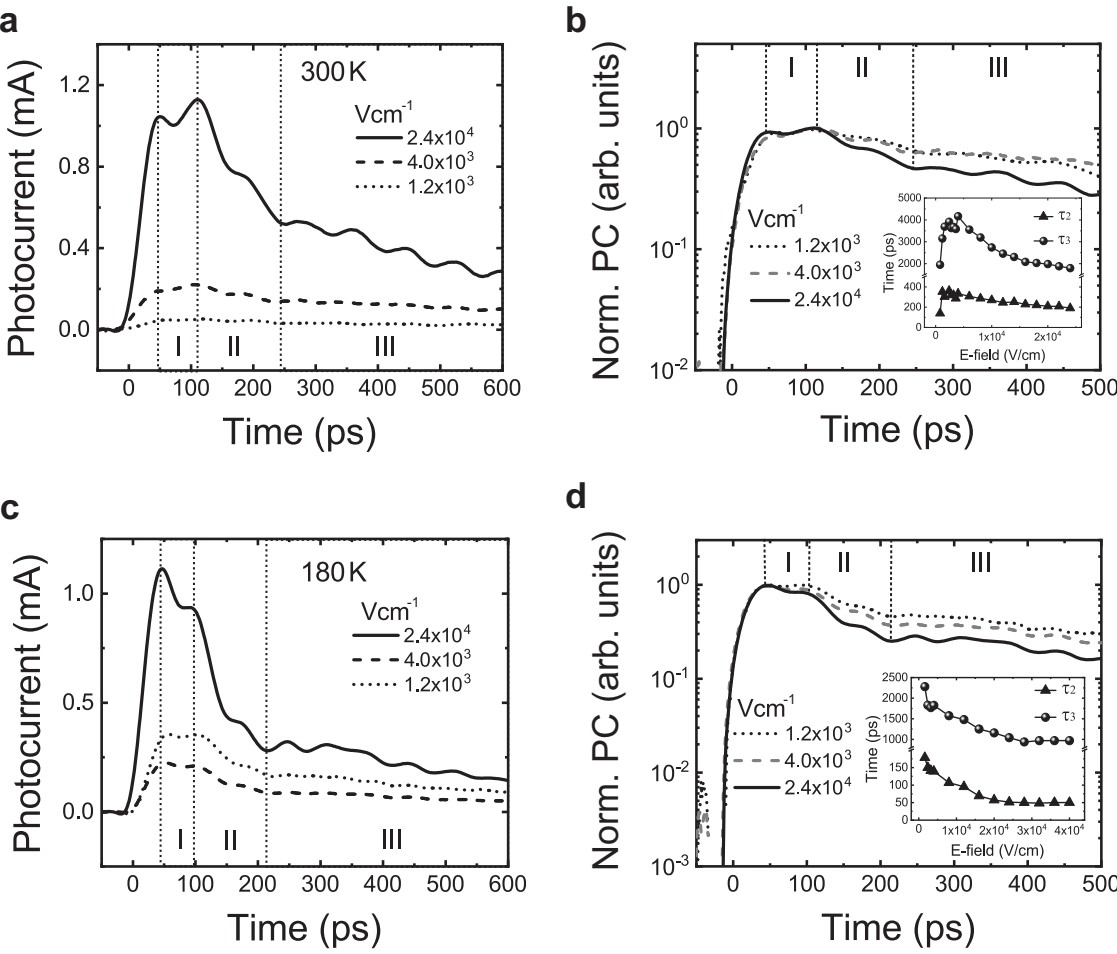

**Fig. 4 Ultrafast photocurrent dependence on electric field. a**, **b** The ultrafast photocurrent and corresponding normalized photocurrent under 300 K. **c**, **d** The ultrafast photocurrent and corresponding normalized photocurrent under 180 K. The insets are the extracted decay times in regions II and III. Note that the oscillations are a result of impedance mismatch, which slightly varies with devices and do not correspond to an internal physical process.

**Table 1 Trap parameters.**

| Decay region | Trap level ($E_a$) (meV) | Trap density (cm$^{-3}$) | Trapping rate (s$^{-1}$) | De-trapping rate (s$^{-1}$) | Capture cross section (cm$^2$) |
|---|---|---|---|---|---|
| I | 1.72 | $3 \times 10^{15}$ | $1.14 \times 10^{11}$ | $3.7 \times 10^{10}$ | $1.02 \times 10^{-16}$ |
| II | 7.02 | $2.6 \times 10^{15}$ | $2.94 \times 10^{10}$ | $2.17 \times 10^{10}$ | $7.37 \times 10^{-17}$ |
| III | 11.17 | $2.3 \times 10^{15}$ | $1.53 \times 10^{9}$ | $1.41 \times 10^{9}$ | $2.60 \times 10^{-18}$ |

carrier de-trapping is reduced. This will lead to a reduction in peak photocurrent at larger external electric fields. To further validate our mechanism, we measure the ultrafast photocurrent dependence with laser fluence to emphasize the trap-assisted transport process (Supplementary Fig. 8).

## Discussion

To quantitively validate the transport mechanisms in regions I, II, and III, and map out the trapped carrier transport in the landscape of energetic and temporal scales, we characterize the trap properties in the framework of the rate dynamics and the PAT model to describe carrier trapping and de-trapping dynamics. Moreover, the PAT model has been used successfully to describe the de-trapping effects of trap states in organic and amorphous semiconductors[50], but to our knowledge has not been incorporated into perovskites, thus far due to the inability to measure ultra-shallow trap levels. In the PAT model, the carrier emission

rate from the trap states can be written as,

$$k(T, \mathbf{F}) = k_0 e^{-\left(\frac{E_a - \alpha F^2}{k_B T}\right)} \tag{1}$$

where, $k(T,\mathbf{F})$ is the carrier emission rate dependent on temperature ($T$) and electric field ($\mathbf{F}$), $k_0$ is the zero-emission rate, $k_B$ is the Boltzmann constant, $E_a$ is the zero field of the trap level, and $\alpha$ is a constant related to the material permittivity. The calculated parameters valid in the 180–300 K region are listed in Table 1, and a detailed analysis can be found in Supplementary Note 3 and Supplementary Table 1.

In region I, the zero field emission rates at 300 K (Supplementary Fig. 10) is most accurate when described for $\mathbf{F}^2$. Fitting of Eq. (1) with photocurrent decay rate indicates the de-trapping rate in ultra-shallow traps is initially in the order of $10^{10}$ s$^{-1}$, and the probability of carrier depopulation to the CB from traps due to phonon interactions (via PAT) is high and reduces slightly as carriers move into shallow traps. This is validated by the increase

in tunneling time for these carriers from ~0.29 to ~0.31 ps (see Supplementary Note 3) due to the increase in trap level and consequently the energy barrier, while they move from traps to the CB in regions I and II, respectively. From Eq. (1), the decay rates are likewise inversely proportional to temperature. Thus, the zero temperature decay rates (Supplementary Fig. 11) reveal trapping rates in each region. The initial trapping rate of $\sim 10^{11}$ s$^{-1}$ is larger than the initial de-trapping rate, which indicates that while most carriers will be trapped in the ultra-shallow traps in region I and are de-trapped into the CB via PAT, some undergo recombination within the traps (trap-assisted recombination). In region II, de-trapping is also prevalent as seen in Fig. 2f and at a similar rate to region I, confirming the PAT behavior in both regions I and II. In region III, the de-trapping rate reduces significantly, indicating limited de-trapping into the CB due to the increase in the barrier potential and tunneling time. For region III shallow traps, the trapping rate is lower as well and as seen in Fig. 2g, constant trap levels of ~11.01 meV indicate carriers moving among these shallow trap states.

In addition to the trapping and de-trapping rates resulting from the PAT transport model, the CCS values, another unique property of trap states, can be derived from the de-trapping rate in all three regions. Supplementary Figure 12 indicates that the CCS radius decreases from $\sim 10^{-8}$ cm (1 Å) in region I to $\sim 10^{-9}$ cm (0.1 Å) in region III. To the best of our knowledge, the reported transport properties (CCS values, trapping rates, and de-trapping rates) for ultra-shallow traps in the ultrafast regions (<1 ns) are the first for inorganic metal halide perovskite thin films and lowest reported trap levels. The product of CCS and trap density that provides a numerical value to indicate the trapping probability, decreases from $\sim 10^{-1}$ cm$^{-1}$ (region I) to $\sim 10^{-3}$ cm$^{-1}$ (region III; i.e., a reduction of trapping probability), and is consistent with results from Shi et al.[45] and Baloch et al.[51], which report reduction in trapping probability with the trap depth. This decreasing trend of trapping probability is consistent with the decreasing in trapping and de-trapping rates in regions I and III from $\sim 10^{11}$ to $\sim 10^{9}$ s$^{-1}$ and $\sim 10^{10}$ to $\sim 10^{9}$ s$^{-1}$, respectively.

Our work has profound implications toward synthesis of highly defect-tolerant bulk perovskite materials, as well as their counterparts, nanostructured perovskites. In addition to organic/inorganic metal halide perovskite synthesis, our work directly bridges the gap between carrier dynamics and device applications in optoelectronics and photonics, establishing a fundamental foundation for characterizing trap dynamics in these high defect-tolerant perovskites with ultrafast temporal resolution and ultrahigh energetic resolution. The traps play a crucial role in carrier transport as described by the framework of the PAT and hopping transport mechanisms. The profound knowledge of the detailed energetics of the shallow traps described by the PAT model indicates that carriers trapped in ultra-shallow trap states tunneling to CB to contribute current rather than carry out trap-assisted recombination. Along with an understanding of the composition and structural origins of the trap states, we can tune and manipulate the trap dynamic properties, such as trapping, de-trapping rates, and their interplay with trap properties, such as capture cross section, trap level, and trap density to fundamentally improve device efficiency and stability performance. This can be done by defect passivation to convert deep traps to ultra-shallow trap through chemical passivation in solution-processed materials[40], where an increase in the ratio between shallow and deep traps will greatly improve device efficiency and response times. Steady-state devices such as solar cells operate under constant light and although they will continuously generate excited carriers, a majority will recombine in deep traps thus reducing efficiency. Thus, ultra-shallow traps identified in this work will enable us to map new trap levels to engineer more efficient carrier transport pathways reducing carrier loss due to recombination and synthesize more efficient materials for these steady-state devices. Integration of ultrafast laser spectroscopy with high-speed optoelectronics will also reveal other subtle "temporal blind spots" in ultrafast carrier dynamics in situ devices, which cannot be achieved by previous all-optical ultrafast laser spectroscopy and steady-state characterization methods.

## Data availability
All data needed to evaluate the conclusions in the paper are present in the paper and/or the Supplementary information.

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

## Acknowledgements
K.K., P.A., and J.G. acknowledge the support from the South Carolina Research Authority (SCRA) and start-up fund from Clemson University. S.S., H.T., and W.N. acknowledge this work was performed, in part, at the Center for Integrated Nano-technologies, an Office of Science User Facility operated for the U.S. Department of Energy (DOE) Office of Science by Los Alamos National Laboratory (Contract 89233218CNA000001) and Sandia National Laboratories (Contract DE-NA-0003525). W.N. acknowledges Los Alamos National Laboratory's LDRD program for support. We also wish to thank Russell Reynolds, Barrett Barker, and Michael Denz at Clemson University for their instrumental support.

## Author contributions
K.K., S.S., E.L., and P.A. performed the experiments. K.K. and S.S. contributed equally to this work. L.C., J.Z., Y.S., Y.Z., Y.B., F.Y., A.M.R., H.T., and M.C.B. were involved in the technical discussion. W.N. and J.G. supervised and designed the project, and wrote the manuscript with K.K. All authors discussed the results and commented on the manuscript.

## Competing interests
The authors declare no competing interests.
