## [Peer Review File · Nature Communications]

REVIEWER COMMENTS

Reviewer #1 (Remarks to the Author):

The paper by K. Kobbekaduwa et al., studied trapped dynamics including trapping, de-trapping, activation energy and capture cross section of traps in a lateral Au/MAPbI₃/Au device using an ultra-fast photocurrent spectroscopy. It was explained based on the results of temperature dependent photocurrent that the trapped carriers undergo a phonon assisted tunneling mechanism and later a hopping transport mechanism along ultra-shallow to shallow trap states ranging from 1.72-11.51 meV. While this is an interesting study there are multiple discussions which need to be clarified before this manuscript can be accepted for publication in Nature Communications.

1) The available data in multiple researches show that trap-assisted recombination exist in perovskite devices. I am not clear how such an ultra-shallow trapping (ps) can impact the non-radiative or trap assisted recombination in perovskite devices?

2) I have some concern with regards to the sample design as characterization of lateral electrode devices provide surface properties rather than bulk. Also, the electric field (4000V/cm) applied in this study is around 10 times smaller than the electric field in a solar cell. I am not sure if such a small electric field can induce ion migration in perovskite devices.

3) Another main concern is that MAPbI₃ undergoes a phase transition at around 160K from tetragonal to orthorhombic structure. This phase transition may affect on the temperature dependent studies and calculation of activation energy and capture cross-section of defects.

4) What is the interplay of charge carriers with ultra-fast trapping events? How charge transport can be affected by these fast trapping?

5) Please clarify how the trap levels are assigned close to the CB and none of them are close the VB?

6) The charge mobility calculated in this study is much smaller than the estimated mobility by other techniques. Usually ultra-shallow defects with fast trapping and de-trapping results in much higher mobility.

7) The authors can provide a better literature study in the introduction and perhaps compare the results with ns- μ s trapping and de-trapping events observed by other methods. For example, much longer trapping and de-trapping events was observed by Musiienko et al. as well as deep defects in the middle of the gap.

1- Musiienko et al., Deciphering the effect of traps on electronic charge transport properties of methylammonium lead tribromide perovskite Sci. Adv. 2020; 6 : eabb6393

2- Musiienko et al., Deep levels, charge transport and mixed conductivity in organometallic halide perovskites, Energy Environ. Sci., 2019, 12, 1413-1425

Reviewer #2 (Remarks to the Author):

This work by K. Kobbekaduwa et al. reports the ultrafast dynamics of trapped carriers in MAPbI₃ thin films by ultrafast photocurrent spectroscopy (UPCS) with a sub-25 ps time resolution. Upon ultrafast laser excitation, the authors proposed that the trapped carriers undergo a phonon assisted tunneling (PAT) mechanism and later a hopping transport mechanism along ultra shallow to shallow trap states ranging from 1.72-11.51 meV. Using this method, the authors estimated carrier mobility of ~ 4 cm²/Vs and lifetime of ~ 1 ns. Understanding the shallow trap states in halide perovskites is an important question and there is very limited amount of work reported. I believe this work provides important insights regarding the transport behavior in the trap states. I can recommend publication in nature communications after the author addressing the following questions.

1. Within time scale of 1 ns, a mobility of $\sim 4 \text{ cm}^2/\text{Vs}$ and a electrical field $\sim 10^4 \text{ V/cm}$, the charge carries probably only travel $\sim 100 \text{ nm}$, if my estimation is correct. That means this method can only probe the defects within 100 nm from the metal contact. So, how about the trap states farther away from the metal contact (deeper inside the film)?
2. In this work, Au is in direct contact with perovskite. This is not the case in many device applications such as solar cells and LEDs, where HTL/ETL/HBL/EBL are used. Will the imperfect metal-perovskite direct contact influence the results?
3. Au is used as the contact electrode, does it accept hole only (or it also collect electrons)? If both types of carriers are collected, is there a way to distinguish them? Another related question, Where is the Fermi level for the perovskite film used in this work?
4. In page 3, line 62-70, the authors mentioned it is difficult to study long-range charge transport using optical spectroscopy. Exciton diffusion in longer ranges have been studied using ultrafast imaging techniques, may be the following works can be mentioned briefly in the introduction (Nature Communications 2020, 11, 664; Nature Mater. 2020, 19, 412-418; etc.)
5. In Figure 1, the perovskite crystal structure shows two colors for the octahedron (purple and blue), is this correct? It looks like a double perovskite structure (e.g., Ag-Bi) as the normal perovskite should be illustrated with only one color. Please check it to avoid any confusion.

Reviewer #3 (Remarks to the Author):

The authors perform a detailed study to determine some of the material properties of MAPbI₃ films. Generally I think the authors have carried out a high quality study, though there are some critical questions that should be addressed, as below.

1. Fig S2 shows quite different behaviour compared to fig. 2. So how general are the results? Are the actual values supposed to be of significance, or more the mechanism of transport? Also, the extracted lifetime of 1ns is very low compared to typical literature values for perovskite films, which also raises the question of how representative the results are.
2. How can you distinguish between shallow electron traps and shallow hole traps?
3. Where do the values in fig. 3 come from (1.72, 4.51, 7.02meV)? It is not at all clear how they are extracted from the data given the activation energies in fig. 2 vary continuously with time.
4. The authors calculate the mobility but it seems it is actually the sum of the electron and hole mobilities that is being calculated. More information (in the supplementary section) would be useful.
5. The surface of the perovskite area being probed is unpassivated. To what extent could surface defects impact on the results?
6. The authors say that the anomalous behaviour at high fields is dues to ion movement. It is not clear why there would be a threshold value for the field beyond which ion movement would occur, so this statement needs justification. Indeed, unless the electric field is pulsed (this should be clarified) then the ions would be expected to screen the field in all cases.
7. The practical importance of the results is not made very clear. What is the implication of the determined trap energies and the mechanisms of decay for devices operating at steady-state?

Reviewer 1 general comment:

While this is an interesting study there are multiple discussions which need to be clarified before this manuscript can be accepted for publication in Nature Communications.

Comment 1: The available data in multiple research show that trap-assisted recombination exists in perovskite devices. I am not clear how such an ultra-shallow trapping (ps) can impact the non-radiative or trap assisted recombination in perovskite devices?

Response: The reviewer makes an excellent point, as multiple research groups have focused on the trap-assisted recombination (eg. Zhang, Z.-Y. et al., Wetzelaer, G. A. H. et al.). Typically, trap-assisted recombination processes occur in deeper traps and is easier to resolve as their temporal range are longer than ~10 ns, while ultrafast shallow trapping (less than 1 ns) cannot be resolved by their characterization approach. As a result, one of the novelties of this work is the characterization of traps in the ultrafast region less than ~1 ns, as reflected from the manuscript title. Meanwhile, the ultra-shallow trapping mechanism has a profound impact on solar cells as they can improve the Voc from carriers that can be extracted prior to trap assisted recombination. Hence, ultra-shallow trapping allows us to mitigate carrier non-radiative or trap assisted recombination.

In the manuscript

1. We add references, page 8, line 173,
 - (1) Zhang, Z.-Y. et al. The role of trap-assisted recombination in luminescent properties of organometal halide CH₃NH₃PbBr₃ perovskite films and quantum dots. Sci. Rep. 6, 1–7 (2016).
 - (2) 39. Wetzelaer, G. A. H. et al. Trap-assisted non-radiative recombination in organic–inorganic perovskite solar cells. Adv. Mater. 27, 1837–1841 (2015).
2. Page 8, lines 170-176, we add **“The existence of ultra-shallow traps leads to ultrafast trapping events. These traps aid in faster transport of charge carriers due to their proximity to the CBM, ease of de-trapping into the CBM and lowering of the dominant non-radiative recombination processes in organic lead halide perovskites^{38,39} which will result in typically lower carrier transport times and higher device efficiency compared to deeper and slower trapping events. In solar cells this can lead to an increase of the open-circuit voltage (Voc) from carriers that can be extracted before trap assisted recombination occurs.”**
3. Page 14-15, lines 288-295, we add **“This can be done by the way of defect engineering in solution processed materials, where an increase in the ratio between shallow and deep traps will greatly improve device efficiency and response times. Steady state devices such as solar cells operate under constant light and although they will continuously generate excited carriers, a majority will recombine in deep traps thus, reducing efficiency. Thus, ultra-shallow traps identified in this work will enable us to map new trap levels to engineer more efficient carrier transport pathways reducing carrier loss due to recombination and synthesize more efficient materials for these steady state devices.”**

Comment 2: I have some concern with regards to the sample design as characterization of lateral electrode devices provide surface properties rather than bulk. Also, the electric field (4000V/cm) applied in this study is around 10 times smaller than the electric field in a solar cell. I am not sure if such a small electric field can induce ion migration in perovskite devices.

Response: Indeed, although the device electrode geometry is co-planar(lateral) rather than the sandwich geometry in the case of solar cell, the photogenerated carrier transport properties studied are from the bulk of the film because the photon absorption depth of 400 nm light is ~ 200 nm, which is much deeper than the depth of surface states, as illustrated in the schematic figure below.

As indicated in the Fig. 4, the electrical field threshold for ion migration to affect detrapping is ~ 4000 V/cm and extend to the studied range 10^5 V/cm, which is in the same range of solar cell built-in electrical field. Also, the reference from Wang, Zhou and Luo (Wang, H., Zhou, M. & Luo, H. Electric-Field-Induced Dynamic Electronic Junctions in Hybrid Organic–Inorganic Perovskites for Optoelectronic Applications. ACS omega 3, 1445–1450 (2018).) indicates the ion migration occurs at even lower electric fields in a similar device setup for perovskites.

In the manuscript:

1. Page 11-12, line 219-224, we add “**Although ion migration is ever present with the application of an external electric field, Fig 4b and d indicate the screening effects due to ion migration will begin to influence the free flow of charge carriers (photocurrent) when the external electric field is larger than 4×10^3 V/cm as the accumulation of ions on the perovskite-gold interface will result in the formation of an internal electric field that directionally opposes the externally applied electric field, reducing the flow of charge carriers.**”

Comment 3: Another main concern is that MAPbI_3 undergoes a phase transition at around 160K from tetragonal to orthorhombic structure. This phase transition may affect on the temperature dependent studies and calculation of activation energy and capture cross-section of defects.

Response: Yes, we agree with reviewer’s comment. MAPbI_3 does undergo a phase change at around 160K hence our study is concentrated on temperatures above 180 K. The activation energies and capture cross section are only relevant to temperatures greater than 180 K.

In the manuscript:

1. Page 12 and line 243-244, we add **“The calculated parameters valid in the 180 K – 300 K region are listed in table 1, and a detailed analysis can be found in SI.”**

Comment 4: What is the interplay of charge carriers with ultra-fast trapping events? How charge transport can be affected by these fast trapping?

Response: we have addressed this in a similar comment from reviewer #1, comment #1.

Comment 5: Please clarify how the trap levels are assigned close to the CB and none of them are close the VB?

Response: The photocurrent from the UPCS approach is a sum of electron and hole transport dynamics, therefore, we are not able to distinguish between electron or hole. In the manuscript, we use electron as an example (hence, the reason for trap levels to be close to the CB), same trapping property and mechanism apply to holes as well.

Comment 6: The charge mobility calculated in this study is much smaller than the estimated mobility by other techniques. Usually, ultra-shallow defects with fast trapping and de-trapping results in much higher mobility.

Response: The calculated mobility is the lowest estimated value for an assumed quantum yield of 100 %. For real quantum yields (< 100 %) the mobility is much higher.

In the manuscript, we highlight the mobility is the minimum estimate mobility, in page 5 line 116-117, we add **“minimum estimate of ~ 4 cm²/Vs for a quantum yield of 100 %”** and in page 10 line 202, we add **“lowest estimate”**. It is further highlighted in the SI (page 11, line 103-104; **“Note: the mobility calculation is the lower estimated value because we assume the quantum yield is 100 %.”**)

Comment 7: The authors can provide a better literature study in the introduction and perhaps compare the results with ns-μs trapping and de-trapping events observed by other methods. For example, much longer trapping and de-trapping events was observed by Musiienko et al. as well as deep defects in the middle of the gap.

Response: To make a link between our ultrafast work with ns-μs trapping and de-trapping observed by other approaches, we compare those longer trapping techniques with ours.

In the manuscript this is highlighted in page 4, line 87-94 where we add **“Hence, these techniques are ideal for characterizing deeper trap levels to which carriers take an extended time period to decay such as those existing in the bulk of single crystalline perovskites, as described by Musiienko et al.^{23,24} However, for accurate measurements these techniques require large device cross sections resulting in greater device RC constants that lead to device rise times in the order of ~10 ns (see SI). This leads to the omission of the ps-ns region in the photocurrent decay**

thus, these techniques are “blind” to shallow trap dynamics which happen near the edge of the conduction band and occur in this ps-ns time scale or even less.”

We add references:

1. Musiienko, A. et al. Deep levels, charge transport and mixed conductivity in organometallic halide perovskites. *Energy Environ. Sci.* 12, 1413–1425 (2019).
2. Musiienko, A. et al. Deciphering the effect of traps on electronic charge transport properties of methylammonium lead tribromide perovskite. *Sci. Adv.* 6, eabb6393 (2020).

Reviewer 2 general comment:

I believe this work provides important insights regarding the transport behavior in the trap states. I can recommend publication in nature communications after the author addressing the following questions.

Comment 1: Within time scale of 1 ns, a mobility of $\sim 4 \text{ cm}^2/\text{Vs}$ and an electrical field $\sim 10^4 \text{ V/cm}$, the charge carriers probably only travel $\sim 100 \text{ nm}$ if my estimation is correct. That means this method can only probe the defects within 100 nm from the metal contact. So, how about the trap states farther away from the metal contact (deeper inside the film)?

Response: The travel length calculation by the reviewer is correct. Carriers are excited from the whole device area as the laser illuminates whole of the device area including the bulk film and parts of the perovskite/Au interfaces as demonstrated by the schematic figure below, wherein the bulk film has a length of $25 \mu\text{m}$, in comparison to the less than $1 \mu\text{m}$ interface length. As mentioned in the response to comment #2 from reviewer #1 the current is a flow of carriers, as a result, the photocurrent contribution is from deep inside the film, or the bulk, instead of the interface.

Comment 2: In this work, Au is in direct contact with perovskite. This is not the case in many device applications such as solar cells and LEDs, where HTL/ETL/HBL/EBL are used. Will the imperfect metal-perovskite direct contact influence the results?

Response: We totally agree with the review comment. To avoid perovskite degradation due to contacts such as silver, (Svanstrom, S. et al. Degradation Mechanism of Silver Metal Deposited on Lead Halide Perovskites. *ACS Appl. Mater. Interfaces* 12, 7212–7221 (2020).), in particular, we use Au electrodes. As discussed, in comment #1, the photocurrent contribution is mainly from the bulk film between electrodes hence, the effect of the interface can be neglected. We have also included the dark IV curve in the SI (page 3, Fig S2) to show the ohmic nature of contacts at lower voltages.

In manuscript:

1. page 6, line 132-136, we add “**The ultrafast photocurrent generated upon an ultrafast laser excitation (400 nm, 130 fs pulse laser) is collected by a high bandwidth sampling oscilloscope, while the laser illuminates whole device including bulk perovskite film and Au/perovskite interfaces (shown in Fig. S8). As a results, charge carriers which contribute to the ultrafast photocurrent is mainly from the bulk of the film.**”
2. Page 6, line 139, we add “**The ohmic nature of the contacts is shown by the dark IV given in Fig S2.**”

Comment 3: Au is used as the contact electrode; does it accept hole only (or it also collect electrons)? If both types of carriers are collected, is there a way to distinguish them? Another related question, where is the Fermi level for the perovskite film used in this work?

Response: We collect both types of carriers as photocurrent but currently they cannot be distinguished from each other. The fermi level for the film is near to the CB around ~100 meV below the CB edge.

Comment 4: In page 3, line 62-70, the authors mentioned it is difficult to study long-range charge transport using optical spectroscopy. Exciton diffusion in longer ranges have been studied using ultrafast imaging techniques, may be the following works can be mentioned briefly in the introduction (Nature Communications 2020, 11, 664; Nature Mater. 2020, 19, 412-418; etc.)

Response: Although excitons diffuse in longer ranges, they carry zero current because they are neutral particles. As a result, optical spectroscopies are the ideal and powerful approach to probe their diffusion dynamics through studying the emitted photons when electron-hole recombination occurs. To place our work in a wider background, In the manuscript:

1. Page 3 line 73, we add the references *Nature Communications 2020, 11, 664; Nature Mater. 2020, 19, 412-418;*

2. Page 3-4, line 72-77, we add, “**While new transient absorption imaging techniques^{19,20} have proven to be powerful tools that can probe long range diffusion dynamics through analysis of photon emission at electron-hole recombination, the long-range measurements are primarily focused on excitons which are charge neutral electron-hole pairs that do not carry current, thus aren’t useful in charge carrier transport studies required for in-situ devices.**”

Comment 5: In Figure 1, the perovskite crystal structure shows two colors for the octahedron (purple and blue), is this correct? It looks like a double perovskite structure (e.g., Ag-Bi) as the normal perovskite should be illustrated with only one color. Please check it to avoid any confusion.

Response: Thank you. Noted and changed in Fig. 1

Reviewer 3 general comment:

The authors perform a detailed study to determine some of the material properties of MAPbI₃ films. Generally, I think the authors have carried out a high-quality study, though there are some critical questions that should be addressed, as below.

Comment 1: Fig S2 shows quite different behavior compared to fig. 2. So how general are the results? Are the actual values supposed to be of significance, or more the mechanism of transport? Also, the extracted lifetime of 1ns is very low compared to typical literature values for perovskite films, which also raises the question of how representative the results are.

Response: Both fig S2 and fig 2 represent temperature dependence of two similar devices. Both are normalized curves, but the normalization point is different hence the appearance of different behavior, but their behavior is perfectly indicated in the similarity of the temperature dependence of the decay times shown in the insets. The values are more representative of the transport mechanism and represent a range. The lifetime shown is an ultrafast photocurrent lifetime and not the general photoluminescence lifetime that is given as the lifetime of perovskite in many literatures. Our main goal is to define charge carrier dynamics for in-situ devices thus, carrier lifetimes would be much lower due to the applied external electric field which increases carrier drift and recombination rates.

In the SI, page 3, Fig. S1 caption, we add “**This value is for photocurrent lifetime in-situ devices under an external electric field and is different from ‘traditional’ lifetimes obtained from steady state photoluminescence measurements which represent ex-situ materials.**”

Comment 2: How can you distinguish between shallow electron traps and shallow hole traps?

Response: As addressed in reviewer #2, comment #3, we are unable to distinguish between electrons and holes because our technique collects both types of particles. From our activation energy calculations, we can identify the position of the trap with

respect to the band edge. Our transport mechanism is described in page 9 Fig 3. (caption) for electron motion by assumption. Thus, we have assumed the traps to be shallow electron traps here, but a similar mechanism can be used to describe hole motion by assuming they are shallow hole traps.

Comment 3: Where do the values in fig. 3 come from (1.72, 4.51, 7.02meV)? It is not at all clear how they are extracted from the data given the activation energies in fig. 2 vary continuously with time.

Response: Fig. 3 is primarily used to highlight the transport model observed from the data. The values in fig. 3 are typical values in that specific region calculated from slopes of fig. 2b, c and d using Arrhenius relation. As pointed out the activation energies do vary in regions I and II thus, to make the transport process clear, we have included only certain levels when in reality they are continuous levels as pointed by the reviewer.

In the manuscript, page 9, Fig. 3 caption, we add **“Note: To clarify the transport process and for convenience only some of the typical states existing within each region are shown. Whereas in the actual case trap levels in the regions between 0-2000 ps are continuous as seen in fig. 2.”**

Comment 4: The authors calculate the mobility but it seems it is actually the sum of the electron and hole mobilities that is being calculated. More information (in the supplementary section) would be useful.

Response: The reviewer is correct. The calculated mobility is the sum of electrons and holes.

In the SI, page 10, line 79-81, we add **“The mobility calculated in this work is essentially a combination of both electrons (μ_e) and holes (μ_h) since we cannot identify each separately using our technique.”** Also, we define the calculated mobility as an effective mobility value with equation in line 81, ($\mu_{eff} = \mu_e + \mu_h$)

Comment 5: The surface of the perovskite area being probed is unpassivated. To what extent could surface defects impact on the results?

Response: As addressed in the comment #2 from reviewer #1, although the surface is not passivated, carrier transport and photogenerated current is generated from the bulk of the film. Therefore, the surface effects can be neglected.

Comment 6: The authors say that the anomalous behavior at high fields is due to ion movement. It is not clear why there would be a threshold value for the field beyond which ion movement would occur, so this statement needs justification. Indeed, unless the electric field is pulsed (this should be clarified) then the ions would be expected to screen the field in all cases.

Response: This is addressed in the response to comment #2 from reviewer #1

Comment 7: The practical importance of the results is not made very clear. What is the implication of the determined trap energies and the mechanisms of decay for devices operating at steady-state?

Response: In manuscript page 14-15 line 288-295, we bridge the gap between ultrafast trap mechanism with that of steady state. We add “**This can be done by the way of defect engineering in solution processed materials, where an increase in the ratio between shallow and deep traps will greatly improve device efficiency and response times. Steady state devices such as solar cells operate under constant light and although they will continuously generate excited carriers, a majority will recombine in deep traps thus, reducing efficiency. Thus, ultra-shallow traps identified in this work will enable us to map new trap levels to engineer more efficient carrier transport pathways reducing carrier loss due to recombination and synthesize more efficient materials for these steady state devices.**”

REVIEWER COMMENTS

Reviewer #1 (Remarks to the Author):

The authors have addressed my comments and concerns. The manuscript is ready for publications in the current format.

Reviewer #2 (Remarks to the Author):

The authors have addressed my technical questions and concerns. One minor issue is that the perovskite crystal structure illustration in Fig. 1 is not revised. This should be updated in the final accepted version.

Reviewer #3 (Remarks to the Author):

The authors have addressed most of the concerns raised. However, in my mind two are outstanding and need further clarification, as below.

Response to comment 1 by R1: The authors state "Meanwhile, the ultra- shallow trapping mechanism has a profound impact on solar cells as they can improve the Voc from carriers that can be extracted prior to trap assisted recombination. Hence, ultra-shallow trapping allows us to mitigate carrier non-radiative or trap assisted recombination.

It is not clear what the proposed mechanism is by which shallow traps improve Voc. It is important to be clear about what comparison is being made here. It is not disputed that shallow traps are far less recombination active than deep traps, so it is better to have shallow traps than deep traps. Often, deep traps can be converted to shallow traps through various chemical passivation strategies, so indeed the density of one can increase at the expense of the other. That is basically just saying that defect passivation (which converts deep traps to shallow traps or to traps whose energy levels lie outside the bandgap) makes sense.

But I am not sure if that is what the authors are trying to say, or whether they are arguing that a higher density of shallow traps in a material in which the density of deep traps is constant would be beneficial.

At any rate, the larger question remains - how does knowledge of the detailed energetics of the shallow traps and the trapping mechanism help in improving the material or in predicting its performance?

In response to comment 2 by R1, the authors say "Indeed, although the device electrode geometry is co-planar(lateral) rather than the sandwich geometry in the case of solar cell, the photogenerated carrier transport properties studied are from the bulk of the film because the photon absorption depth of 400 nm light is ~ 200 nm, which is much deeper than the depth of surface states, as illustrated in the schematic figure below"

This argument makes little sense.

i) An absorption depth of 200nm doesn't mean carriers are generated at 200nm depth. Rather, the absorption follows the Beer-Lambert law which means most photogeneration is at the surface.
ii) If the carrier diffusion length is greater than the thickness of the layer, then carriers can diffuse to the surface to recombine.

Therefore, I think the concern about the influence of surface states has not been addressed yet.

Review 1 general comment

The authors have addressed my comments and concerns. The manuscript is ready for publications in the current format.

Reviewer 2 general comment

The authors have addressed my technical questions and concerns. One minor issue is that the perovskite crystal structure illustration in Fig. 1 is not revised. This should be updated in the final accepted version.

Response: Figure 1 has been revised as per below, in the main manuscript as well.

Reviewer 3 general comment

The authors have addressed most of the concerns raised. However, in my mind two are outstanding and need further clarification,

Comment 1:

Response to comment 1 by R1: The authors state “Meanwhile, the ultra- shallow trapping mechanism has a profound impact on solar cells as they can improve the V_{oc} from carriers that can be extracted prior to trap assisted recombination. Hence, ultra-shallow trapping allows us to mitigate carrier non-radiative or trap assisted recombination.

It is not clear what the proposed mechanism is by which shallow traps improve V_{oc} . It is important to be clear about what comparison is being made here. It is not disputed that shallow traps are far less recombination active than deep traps, so it is better to have shallow traps than deep traps. Often, deep traps can be converted to shallow traps through various chemical passivation strategies, so indeed the density of one can increase at the expense of the other. That is basically just saying that defect passivation (which converts deep traps to shallow traps or to traps whose energy levels lie outside the bandgap) makes sense.

But I am not sure if that is what the authors are trying to say, or whether they are arguing that a higher density of shallow traps in a material in which the density of deep traps is constant would be beneficial.

At any rate, the larger question remains - how does knowledge of the detailed energetics of the shallow traps and the trapping mechanism help in improving the material or in predicting its performance?

Response: We totally agree with the review. For the solar cell efficiency, the critical parameters are trap level and density. The profound knowledge of the detailed energetics of the shallow traps described by the phonon assisted tunneling transport model, (which is uniquely identified in this work having ultrafast time resolution and ultrahigh energetic resolution features) indicates that carriers have more opportunity to pop back to conduction band (hole to valence band) and less recombination events while keeping same carrier density. In contrast, in deep trap states, the recombination rate is higher to reduce carrier density. Therefore, as the reviewer suggest, converting deep traps to shallow traps by chemical passivation would be beneficial to the device efficiency since current density is proportional to the carrier density.

In the manuscript:

1. Page 15 line 287-289, we add *“The profound knowledge of the detailed energetics of the shallow traps described by the PAT model indicates that carriers trapped in ultra-shallow trap states tunneling to conduction band to contribute current rather than carry out trap assisted recombination.”* and line 293-295, we add *“This can be optimized by defect passivation to convert deep traps to ultra-shallow trap through chemical passivation in solution processed materials.”*
2. Page 8 line 177, we add reference *‘Fu, L. et al. Defect passivation strategies in perovskites for an enhanced photovoltaic performance. Energy Environ. Sci. 13, 4017–4056 (2020).’*

Comment 2:

In response to comment 2 by R1, the authors say *“Indeed, although the device electrode geometry is co-planar(lateral) rather than the sandwich geometry in the case of solar cell, the photogenerated carrier transport properties studied are from the bulk of the film because the photon absorption depth of 400 nm light is ~ 200 nm, which is much deeper than the depth of surface states, as illustrated in the schematic figure below.”*

This argument makes little sense.

- i) An absorption depth of 200nm doesn't mean carriers are generated at 200nm depth. Rather, the absorption follows the Beer-Lambert law which means most photogeneration is at the surface.
- ii) If the carrier diffusion length is greater than the thickness of the layer, then carriers can diffuse to the surface to recombine.

Therefore, I think the concern about the influence of surface states has not been addressed yet.

Response: We totally agree the reviewer’s concern of surface states. To confirm the photocurrent contribution is from the surface states or not, we also carry out the ultrafast photocurrent measurements wherein laser illumination is from the film bottom surface side through the quartz substrate. The photocurrent demonstrates same decay behavior, as shown in Fig. S3. This is a great indication that the photocurrent contribution is from bulk, not the surface since the quartz surface side has more surface states/defect states, leading

to more recombination events (the indication is much faster decay than that of top surface side illumination).

In the manuscript, Page 6 Line 136-138, we add; *“This is further demonstrated by the similar photocurrent decay trend when laser illuminates from either bottom or top side of the film, while the interface between perovskite film and substrate surface tends to have more surface states. (Fig. S3).”*

In the SI, we add Fig. S3 to indicate the same photocurrent decay trend when laser illuminate from both side of surface film, respectively.

We sincerely hope that the experimental result and reference address the reviewer’s concerns.

REVIEWERS' COMMENTS

Reviewer #3 (Remarks to the Author):

No further comments.

Reviewer 3 (Remarks to the Author): No further comments.

Response: Thanks a lot for the brief comment from reviewer 3. We keep the manuscript as is with no changes, except the response to the editorial requests.